# The Impact of Intolerance of Uncertainty on Negative Emotions in COVID-19: Mediation by Pandemic-Focused Time and Moderation by Perceived Efficacy

**DOI:** 10.3390/ijerph18084189

**Published:** 2021-04-15

**Authors:** Weine Dai, Guangteng Meng, Ya Zheng, Qi Li, Bibing Dai, Xun Liu

**Affiliations:** 1CAS Key Laboratory of Behavioral Science, Institute of Psychology, Chinese Academy of Sciences, Beijing 100101, China; daiweine15@mails.ucas.edu.cn (W.D.); menggt@psych.ac.cn (G.M.); liux@psych.ac.cn (X.L.); 2CFIN and PET Center, Aarhus University, 8200 Aarhus N, Denmark; 3Sino-Danish College, University of Chinese Academy of Sciences, Beijing 101408, China; 4Sino-Danish Center for Education and Research, Beijing 101408, China; 5Department of Psychology, University of Chinese Academy of Sciences, Beijing 100101, China; 6Department of Psychology, Dalian Medical University, Dalian 116044, China; zhengya@dmu.edu.cn; 7Beijing Key Laboratory of Learning and Cognition, Department of Psychology, Capital Normal University, Beijing 100048, China; 8Department of Psychiatry and Psychology, School of Basic Medical Sciences, Tianjin Medical University, Tianjin 300070, China

**Keywords:** COVID-19, pandemic, intolerance of uncertainty, negative emotions, pandemic-focused time, perceived efficacy

## Abstract

The COVID-19 global pandemic has resulted in a large number of people suffering from emotional problems. However, the mechanisms by which intolerance of uncertainty (IU) affects negative emotions during the COVID-19 pandemic remain unclear. This study aimed to explore the mediating role of pandemic-focused time and the moderating role of perceived efficacy in the association between IU and negative emotions during the COVID-19 pandemic based on the uncertainty-time-efficacy-emotion model (UTEE). 1131 participants were recruited to complete measures of COVID-19 IU, pandemic-focused time, perceived efficacy, negative emotions and demographic variables during the COVID-19 pandemic. The results showed that COVID-19 IU was significantly and positively associated with negative emotions, and this link could be mediated by pandemic-focused time. Moreover, the direct effect of COVID-19 IU on negative emotions was moderated by perceived efficacy. Specifically, the direct effect of COVID-19 IU on negative emotions was much stronger for individuals with lower levels of perceived efficacy. The current study further extended the previous integrative uncertainty tolerance model. Furthermore, the study suggested that policy makers and mental health professionals should reduce the general public’s negative emotions during the pandemic through effective interventions such as adjusting COVID-19 IU, shortening pandemic-focused time and enhancing perceived efficacy.

## 1. Introduction

Coronavirus disease (COVID-19) is an extremely severe and highly contagious respiratory disease that has erupted in over 220 countries and territories, with 119.22 million confirmed cases and more than 2.64 million deaths according to a weekly report from the World Health Organization, as of 14 March 2021 [1]. Information about the threat and uncertainty of COVID-19 could not only draw people’s attention to the pandemic and increase their pandemic prevention behavior [2] but could also cause public panic, harm people’s mental health [3] and exacerbate various social problems [4,5,6]. Therefore, it is necessary to study the mechanism by which the uncertainty of pandemic information impacts individual negative emotions to provide effective early warnings and interventions for individual negative emotions.

The integrative uncertainty tolerance model [7] suggested that the perception of the uncertainty of external stimuli could trigger a variety of reappraisals/reactions, such as emotional reactions including worry, fear, and disgust. The model also indicated that individual characteristics, such as self-efficacy, were important moderators of the influence of uncertainty perception on emotional reappraisal/response. Based on this framework, the current study proposed an uncertainty-time-efficacy-emotion model (UTEE) to elucidate emotional responses during the COVID-19 pandemic. In this model, the intolerance perception of the uncertainty of COVID-19 information was considered to be an extension of the uncertainty tolerance model, the individual’s emotional responses were considered to be reappraisals/reactions, and perceived efficacy was the moderator. Additionally, pandemic-focused time, as the index of information seeking, was considered to be a mediator between the perception of pandemic-related uncertainty and negative emotions. An important issue that should be explored is how COVID-19-related intolerance of uncertainty (COVID-19 IU) can affect people’s negative emotions.

### 1.1. COVID-19 IU and Negative Emotions

Intolerance of uncertainty (IU) refers to a dispositional incapacity to endure the negative responses caused by the perception of uncertainty due to a lack of key or sufficient information [8]. The level of IU was positively associated with the level of negative emotions, such as fear [9], anxiety [10] and depression [11]. Individual differences in IU have been widely studied in clinical research and public health crises. In clinical studies, IU has been identified as a transdiagnostic factor for emotional disorders, and IU levels were higher in patients with obsessive-compulsive disorder or/and generalized anxiety disorder than in healthy controls [8,12,13]. During the H1N1 health crisis, individuals with higher IU scores showed higher levels of H1N1-related anxiety and were more likely to perceive the pandemic as threatening [14]. The COVID-19 pandemic involves many uncertainties, including unknown treatment, pathological characteristics, and governance policies. In addition, COVID-19 has thrown a host of issues into uncertainty, including school delivery, economic development, and career planning [15,16]. The perception of these serious uncertainties also caused serious emotional problems, such as fear, anxiety and depression [9,17,18]. Understanding individual differences in COVID-19 IU and its impact on negative emotions during the COVID-19 pandemic is instructive for prevention and health interventions.

### 1.2. Pandemic-Focused Time as a Mediator

Pandemic-focused time reflects the behavior of seeking information, which is one type of behavioral responses of individuals to IU [7,19,20]. The COVID-19 pandemic has caused some individuals to spend excessive time focusing on the event to reduce this feeling of uncertainty by seeking relevant information. However, this excessive focus time implied deep involvement in vital negative events, which might cause vital negative emotions [21]. Recent studies’ findings supported that pandemic-focused time was associated with the risk of anxiety, depressive symptoms and insomnia symptoms during the COVID-19 out-break [18,22,23]. In addition, excessive time focusing on the COVID-19 pandemic was accompanied by a massive “infodemic”, according to the situation report from the World Health Organization [24]. The perceptual overload from the infodemic has increased the level of negative emotions [25]. Fortunately, pandemic-focused time, as a behavioral response of individuals to COVID-19 IU, is easy to measure, easy to observe and controllable, especially in the case of mental health interventions. Therefore, it is necessary to study the role of pandemic-focused time in the influence of COVID-19 IU on negative emotions.

### 1.3. Perceived Efficacy as a Moderator

Perceived efficacy played an important role in the self-regulation of negative emotion during uncertain situations. Faced with uncertain events, people are not always passive perceivers but can actively adjust. In pandemics, perceived efficacy refers to individuals’ confidence that performing preventive measures can effectively alleviate the pandemic and individuals’ belief in their ability to effectively cope with a health threat [26,27]. According to the extended parallel process model, when people perceive a high threat upon exposure to pandemic information, they will experience fear, which motivates perceived efficacy to reduce negative emotion by danger control or fear control [28,29]. If people perceive enough efficacy to avert the threat, they will have an adaptive response; otherwise, they will have a maladaptive response (e.g., avoidance, denial) [28,29]. For example, during the H1N1 pandemic, individuals with a high level of efficacy had a high intention to follow pandemic preventive measures for H1N1 [30,31], while individuals with a low level of efficacy obtained less knowledge about H1N1 [32]. However, the moderating effect of perceived efficacy on negative emotion during the COVID-19 pandemic remains unclear.

### 1.4. Current Study

Based on previous studies on the relationships among these variables, we propose a theoretical model to explore how COVID-19 IU influences individuals’ negative emotion during the COVID-19 pandemic (see Figure 1). We propose the following hypotheses: first, COVID-19 IU directly influences negative emotions; second, pandemic-focused time mediates the relationship between COVID-19 IU and negative emotions; third, perceived efficacy moderates the direct effect of COVID-19 IU on negative emotions. Specifically, the direct relationship between COVID-19 IU and negative emotions is much stronger for individuals with lower levels of perceived efficacy.

## 2. Materials and Methods

### 2.1. Participants and Procedures

This web-based cross-sectional design research was approved by the Institutional Review Board of the Institute of Psychology, Chinese Academy of Sciences, and conducted in accordance with the Declaration of Helsinki. All participants were recruited online from 33 provinces of China from 24 February 2020 to 3 March 2020. A total of 1131 participants completed the questionnaires. Data from participants who gave an incorrect response to a question used to detect whether they answered the questionnaire carefully were excluded from the analysis. Finally, data from 1022 participants (90.4%) were entered into the final statistical analyses. Information about the descriptive characteristics of the participants is presented in Table 1.

### 2.2. COVID-19 IU

COVID-19 IU was measured by three items adapted from a well-established intolerance of uncertainty scale [33]: “The uncertainty of the COVID-19 pandemic has seriously affected my life”, “The uncertainty of the COVID-19 pandemic prevents me from working or studying well”, and “It makes me uneasy, anxious, or stressed not having all the pandemic information I need”. These 3 items were assessed on a 7-point Likert scale (from 1 = strongly disagree to 7 = strongly agree). The total score is equal to the sum of the scores of each item, and higher scores reflect a higher level of COVID-19 IU. Cronbach’s α was 0.75 in current study.

### 2.3. Negative Emotions

Fear, anxiety and depression were used to represent negative emotions in the COVID-19 pandemic [34,35,36]. Referring to the Positive Affect and Negative Affect Schedule (PANAS) [37], participants were required to rate the intensities of these three emotions words. To reduce time consumption and increase flexibility in special cases of the COVID-19 pandemic, we used single-item measurement for each negative emotion, which effectively reduces sample bias and improves data quality [38,39]. Additionally, other common self-report scales request participants to report whether they had experienced each of the related symptoms in the past 7 days (e.g., Self-Rating Anxiety Scale and Self-Rating Depression Scale [40,41]) or 14 days (e.g., Generalized Anxiety Disorder-7, Beck Depression Inventory-II [42,43]); hence, we used the median, 10 days, as the period for measuring emotional intensity. Correspondingly, negative emotions were assessed using three questions (See Table A1). Each negative emotion contributes weighs equally in calculating the total score; higher scores reflect higher levels of negative emotions. The Negative Emotion Questionnaire showed good internal consistency, good convergent and discriminant validity during the COVID-19 pandemic in the previous study [44]. In the present study, the Cronbach’s α coefficient for this questionnaire was 0.87 and the scores of three negative emotions were highly correlated with one another (correlation coefficients between 0.668 and 0.699, *ps* < 0.001, see Appendix A).

### 2.4. Pandemic-Focused Time

Pandemic-focused time spend on COVID-19 was measured by asking participants, “How much time do you spend focusing on COVID-19 information per day on average?” The options were as follows: (a) 10 min or less, (b) 11–30 min, (c) 31–60 min, and (d) more than 60 min.

### 2.5. Perceived Efficacy

Perceived efficacy was assessed using four items from a well-established perceived efficacy scale [27] (See Table A1). These 4 items were assessed on a 7-point Likert scale (from 1 = strongly disagree to 7 = strongly agree). The total score is equal to the sum of the scores of each item, and higher scores reflect a higher level of perceived efficacy. Cronbach’s α was 0.67 in current study.

### 2.6. Procedures

Participants completed the survey by scanning the QR code of the questionnaire link. After reading and signing the informed consent form, participants were required to report their demographic data and all of the questionnaires. Only one response from each IP address was allowed. This web-based survey was completely voluntary and noncommercial.

### 2.7. Statistical Analyses

Data were analyzed using SPSS Version 20.0. Descriptive statistics were used to describe the sample characteristics of each factor. Pearson correlation analyses were used to explore associations between factors conforming to the prerequisites for the following analysis. We tested the hypothesized model using the SPSS macro PROCESS (Model 5) [45]. The bias-corrected percentile Bootstrap method was used to test all regression coefficients. Bootstrapping (5000 bootstrap samples) with 95% confidence intervals (CIs) was conducted to test the significance of mediating and moderating effects. The 95% CIs excluding zero indicate that the effects are significant.

## 3. Results

### 3.1. Preliminary Analyses

The means, standard deviations, and correlation matrix of the variables were presented in Table 2. COVID-19 IU was positively correlated with pandemic-focused time (*r* = 0.170, *p* < 0.001), and negative emotions (*r* = 0.443, *p* < 0.001). The association between pandemic-focused time and negative emotions (*r* = 0.140, *p* < 0.001) and the association between perceived efficacy and negative emotions (*r* = −0.209, *p* < 0.001) were significant. However, the association between COVID-19 IU and perceived efficacy (*r* = −0.053, *p* = 0.091) as well as the association between pandemic-focused time and perceived efficacy (*r* = 0.056, *p* = 0.072) were not significant.

### 3.2. Testing for the Proposed Model

The main results of the proposed model consisted of four parts: Model 1, Model 2, the indirect effect analysis and the conditional direct effect analysis (see Table 3). Model 1 was adopted to test the effect of COVID-19 IU on pandemic-focused time. Model 2 was adopted to test the effects of COVID-19 IU, pandemic-focused time and perceived efficacy on negative emotions. The indirect effect analysis examined the mediation effect of pandemic-focused time on the relationship between COVID-19 IU and negative emotions. The conditional direct effect analysis examined the effects of COVID-19 IU on negative emotions at the mean of perceived efficacy as well as plus and minus one standard deviation from the mean of perceived efficacy.

Controlling for gender, age and education background, the results of Model 1 (*F* = 20.372, *r*${}^{2}$ = 0.074, *p* < 0.001) and Model 2 (*F* = 49.417, *r*${}^{2}$ = 0.254, *p* < 0.001) showed that the direct effect of COVID-19 IU on negative emotions was significant (*B* = 0.445, *SE* = 0.028, *t* = 15.653, *p* < 0.001). The positive predictive effect of COVID-19 IU on pandemic-focused time (*B* = 0.036, *t* = 5.793, *p* < 0.001) and the positive predictive effect of pandemic-focused time on negative emotions (*B* = 0.379, *t* = 2.759, *p* = 0.006) were also significant. Moreover, the upper and lower bounds of the bootstrapped 95% CI for the indirect effect of COVID-19 IU on negative emotions through pandemic-focused time did not include 0, indicating that the mediation effect of pandemic-focused time was significant (indirect effect = 0.138, *SE* = 0.006, 95% CI = [0.003, 0.027]). These results suggested that pandemic-focused time played a partial mediating role in the relationship between COVID-19 IU and negative emotions.

In addition, the negative predictive effect of perceived efficacy on negative emotions was significant (*B* = −0.231, *t* = −7.089, *p* < 0.001). Furthermore, the interaction terms of COVID-19 IU and perceived efficacy showed significant effects on negative emotions (*B* = −0.018, *SE* = 0.008, *p* = 0.008). The results of the conditional direct effect analysis showed that the direct effect of COVID-19 IU on negative emotions was significant at each level of perceived efficacy as the upper and lower bounds of the bootstrapped 95% CI excluded 0. Compared to individuals with high perceived efficacy (direct effect = 0.374, *SE* = 0.035, 95% CI = [0.305, 0.443]), COVID-19 IU had a stronger effect in predicting negative emotions among individuals with low perceived efficacy (direct effect = 0.515, *SE* = 0.042, 95% CI = [0.432, 0.598]) (see Figure 2). Thus, the conditional direct effect analysis showed that perceived efficacy buffered the direct effect of COVID-19 IU on negative emotions. Overall, our findings showed that the indirect effect of COVID-19 IU on negative emotions was aggravated through the mediation of pandemic-focused time, as well as the direct effect of COVID-19 IU on negative emotions was alleviated by perceived efficacy.

## 4. Discussion

The worldwide pandemic of COVID-19 has led to severe emotional problems for many people. The current study first used pandemic-focused time as a mediating variable and perceived efficacy as a moderating variable to discuss the mechanism of negative emotions during the COVID-19 pandemic. The results indicated that people who could not tolerate uncertainty had higher levels of negative emotions. COVID-19 IU increased negative emotions by increasing pandemic-focused time. The effect of COVID-19 IU on negative emotions was moderated by perceived efficacy. Specifically, for individuals with a low level of perceived efficacy, the higher their level of COVID-19 IU, the higher their levels of negative emotions were. The current research validated the role of the UTEE model in the COVID-19 pandemic. Simultaneously, this study suggested effectively decreasing COVID-19 IU, reducing pandemic-focused time and enhancing individuals’ perceived efficacy could reduce negative emotions during the COVID-19 pandemic.

### 4.1. Associations between COVID-19 IU and Negative Emotions

We found that COVID-19 IU was positively associated with negative emotions, suggesting that individuals with higher COVID IU levels were more likely to feel fear, anxiety and depression. First, when people face threats, COVID-19-related uncertainty can lead to negative emotions. The COVID-19 pandemic is a huge crisis for human society that is accompanied by extreme threats. Threat-related uncertainty could cause negative emotions, such as fear and anxiety [12]. Second, individuals with high COVID-19 IU tend to interpret uncertain situations as threats, leading to higher levels of negative emotions. Previous studies have shown that uncertainty itself was perceived as a threat, and individuals with high IU showed a negative interpretation bias for uncertain information and were more likely to treat uncertainty as threatening, which was correlated with anxiety symptoms and depressive symptoms [46,47,48]. Third, individuals with high COVID-19 IU are more eager to seek answers to COVID-19 uncertainty, which increased negative emotions. A previous study found that human papillomavirus (HPV)-infected patients with high IU were more likely to seek HPV information but also perceived higher levels of anxiety than those with lower IU [49]. In our study, one of the items for IU measurement was “If I do not see COVID-19-related information or news, I feel nervous or upset”. Individuals with high COVID-19 IU need more information about COVID-19. However, although individuals spend considerable time searching for COVID-19 information, this does not relieve and may even increase negative emotions because COVID-19 information is rarely definitive and changes constantly during the pandemic.

### 4.2. The Mediating Role of Pandemic-Focused Time

Our findings showed that pandemic-focused time played a mediating role between COVID-19 IU and negative emotions. On the one hand, when faced with uncertainty, individuals with high IU spent more time trying to convert uncertainty into certainty than individuals with low IU during the COVID-19 pandemic. Individuals with high COVID-19 IU were more likely to lose control and less likely to tolerate uncertainty than those with low IU. The former paid too much attention to health-related information (including seeking information about health threats) to meet their greater informational needs than those with low IU [19,49,50]. On the other hand, pandemic-focused time was positively associated with negative emotions. Excessive time focused on the COVID-19 pandemic caused a high level of negative emotions. Uncertainty about health threats is often inherent to the disease and treatment trajectories, and no information can provide permanent certainty about one’s health [49]. Due to the specific characteristics of COVID-19, spending excessive time on the COVID-19 pandemic could not reduce uncertainty and might even lead to strong negative emotions due to excessive attention to negative events [21]. Excessive pandemic-focused time also indicated an overload of pandemic-related information, which could potentially overwhelm individuals and lead to extremely strong negative emotions [25].

### 4.3. The Moderating Role of Perceived Efficacy

Perceived efficacy moderated the direct effect of COVID-19 IU on negative emotions. Specifically, compared with individuals with high perceived efficacy, individuals with low perceived efficacy had more difficulty regulating the negative emotions caused by IU. On the one hand, individuals with a high level of perceived efficacy believed that prevention measures for the COVID-19 pandemic were effective. The efficacy of governmental actions to prevent the COVID-19 pandemic was higher for people who were satisfied with their government prevention measures than those from other countries [51]. They maintained a high belief in the power of COVID-19 countermeasures and had few concerns about the future. Therefore, individuals with high perceived efficacy were less affected by their negative emotions caused by IU than those with low perceived efficacy. On the other hand, individuals with high perceived efficacy believed that they had the ability to cope with the COVID-19 pandemic effectively. Perceived self-efficacy was positively correlated with life satisfaction and reappraisal, a cognitive emotion regulation strategy [52]. Thus, individuals with high perceived efficacy could regulate the effect of COVID-19 IU on negative emotions to a proper level.

### 4.4. Implications for COVID-19 Pandemic Management

In our research, COVID-19 IU increased negative emotions. Due to the specific characteristics of COVID-19, the high uncertainty of COVID-19 cannot be permanently resolved in the short term. COVID-19-related uncertainty could lead to negative emotions and endanger people’s mental health, especially for subpopulations who are at increased risk of being affected by the pandemic. Thus, more attention should be paid to monitoring and focusing on individuals with high COVID-19 IU to prevent them from experiencing excessive negative emotions.

In this case, convenient and effective prevention strategies are urgently needed. Our research found that COVID-19 IU, pandemic-focused time and perceived efficacy can regulate the negative emotions caused by the uncertainty of the COVID-19 pandemic. Therefore, we proposed three strategies to alleviate negative emotions during the COVID-19 pandemic. The first strategy is reducing COVID-19 IU to reduce negative emotions. Several therapies can help individuals with high COVID-19 IU to relieve negative emotions by increasing acceptance of uncertainty. For example, cognitive behavioral therapy targeting IU (CBT-IU) can help individuals reappraise uncertainty and provide cognitive modification for their unrealistically positive fantasies of seeking complete certainty and exposure training to uncertainty [13]. The second strategy is controlling pandemic-focused time to reduce negative emotions. Among individuals with a high level of COVID-19 IU, less pandemic-focused time to obtain COVID-19-related information can help alleviate their negative emotions, while among individuals with a low level of COVID-19 IU, more pandemic-focused time to obtain COVID-19-related information better meets their needs. These results were consistent with previous research that found that matching health messages to individual differences could reduce negative psychological outcomes [53]. The last strategy is increasing perceived efficacy to reduce negative emotions. The government’s efforts to enhance detailed pandemic information, positive risk communication, rumor refutation and medical supplies can increase individuals’ perceived efficacy to defend against the COVID-19 pandemic [27,54,55,56]. To increase self-efficacy, individuals can share reliable sources of information with each other and communicate with professional health/medical staff. More dependable sources of efficacy information can lead to a larger change in self-efficacy [57].

### 4.5. Limitations and Prospects

There are several limitations in the present study. First, although the reliability and validity of the variables in our study reached an acceptable or excellent level, there is still room for improvement in the measurement of some variables. Due to the specificity of the COVID-19 pandemic, all of the measurements were completed online. It is necessary to reduce some variable questions to ensure that the subjects can complete the data carefully and that the data can be collected effectively. In the future, more representative items could be selected according to the situation to further improve the reliability and validity of the research without sacrificing the quality of the respondents’ answers. Second, our sample was limited to a general adult population. Although we excluded the interference of age, gender, and education level, it is undeniable that these variables still have implicit biases. Future studies should further consider the role of the above factors in the UTEE model, and more targeted protection should be provided for specific populations, including other age groups, patients with emotional disorders, populations closely associated with the pandemic (e.g., frontline health-care workers, COVID-19 patients and their close contacts, researchers in related fields, etc.), and populations with high levels of education and related disciplinary backgrounds (e.g., medicine, psychology, environmental sciences, etc.). Third, this study was a cross-sectional study, and only correlation between these variables can be determined. A longitudinal design and clinical trials should be used in the future to determine the causal relationship between these variables.

## 5. Conclusions

The current study constructed a UTEE model to elucidate the effects of individual differences in COVID-19 IU on negative emotions during the COVID-19 pandemic. The results showed a positive association between COVID-19 IU and negative emotions, which was mediated by pandemic-focused time. Moreover, the effect of COVID-19 IU on negative emotions was moderated by perceived efficacy. Accordingly, we propose three strategies to alleviate the negative emotions caused by IU during the COVID-19 pandemic, including decreasing individuals’ COVID-19 IU, reducing the time spent on the COVID-19 pandemic and enhancing perceived efficacy in relation to COVID-19.

## Figures and Tables

**Figure 1 ijerph-18-04189-f001:**
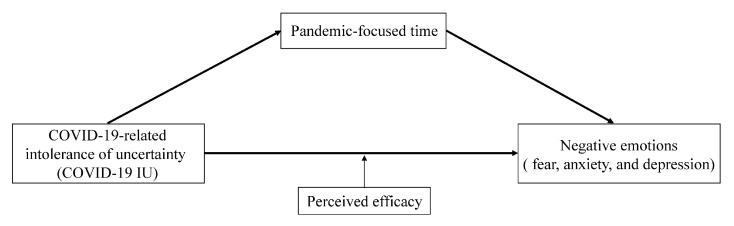
The hypothesized model.

**Figure 2 ijerph-18-04189-f002:**
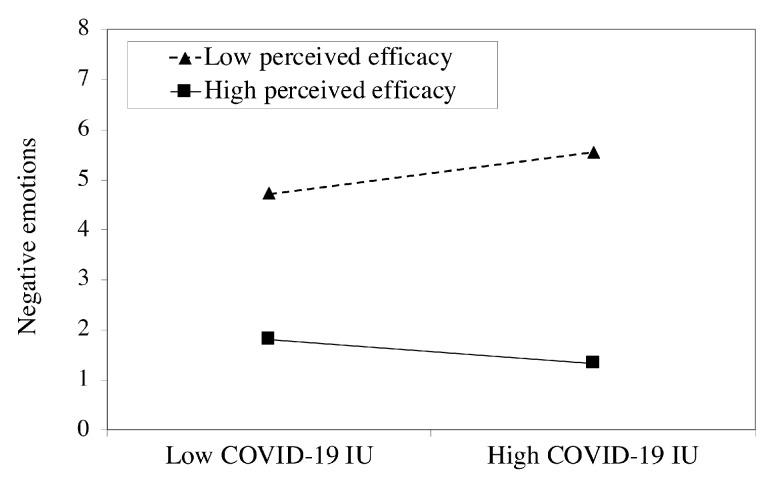
Perceived efficacy moderates the relationship between COVID-19 IU and negative emotions.

**Table 1 ijerph-18-04189-t001:** Demographics of participants.

Variable	Sample Size (Frequency, %)
**Total**	1022 (100.0)
**Gender**	
Male	409 (40.0)
Female	613 (60.0)
**Age**	
18–25	458 (44.8)
26–35	279 (27.3)
36–45	152 (14.9)
46–61	120 (11.7)
Unknown	13 (1.3)
**Education background**	
High school or lower	136 (13.3)
College/technical school	81 (7.9)
University Bachelor’s degree	461 (45.1)
Master’s degree or higher	344 (33.7)
**Career background**	
Student	470 (46.0)
Medical staff	53 (5.2)
Teacher/Lawyer/Civil servant	181 (17.7)
Manager/Office clerk	140 (13.7)
Factory work/Agricultural worker	53 (5.2)
Subcontractor/Service employee	31 (3.0)
Other	94 (9.2)

**Table 2 ijerph-18-04189-t002:** Descriptive statistics and correlations among variables. *N* = 1022.

Variables	M ± SD	1	2	3	4	5	6	7
1. Gender	-	1						
2. Age	30.11 ± 10.13	0.033	1					
3. Education background	-	−0.007	−0.259 ***	1				
4. COVID-19 IU	12.61 ± 4.52	0.073 *	−0.026	−0.021	1			
5. Pandemic-focused time	2.45 ± 0.94	0.037	0.200 ***	−0.003	0.170 ***	1		
6. Perceived efficacy	23.22 ± 3.86	0.057	0.112 ***	−0.076 *	−0.053	0.056	1	
7. Negative emotions	8.75 ± 4.56	−0.084 **	−0.006	−0.015	0.443 ***	0.140 ***	−0.209 ***	1

Note. Gender (0 = female, 1 = male), Education background (1 = High school or lower, 2 = College/technical school, 3 = University Bachelor’s degree, 4 = Master’s degree or higher), Pandemic-focused time (1 = “within 10 min”, 2 = “11–30 min”, 3 = “31–60 min”, 4 = “more than 60 min”). * *p* < 0.05, ** *p* < 0.01, *** *p* < 0.001.

**Table 3 ijerph-18-04189-t003:** Mediation analysis and conditional process analysis.

	*B*	*SE*	*t*	*p*	*Boot LLCI*	*Boot ULCI*
**Model 1**						
*Outcome: Pandemic-focused time*						
Gender	0.034	0.056	0.582	0.561	−0.086	0.149
Age	0.020	0.003	7.009	<0.001	0.015	0.026
Education background	0.055	0.030	1.840	0.066	−0.010	0.120
COVID-19 IU	0.036	0.006	5.973	<0.001	0.023	0.049
**Model 2**						
*Outcome: Negative emotions*						
Gender	−0.996	0.254	−3.927	0.001	−1.505	−0.481
Age	0.004	0.013	0.290	0.772	−0.023	0.029
Education background	−0.126	0.133	−0.948	0.344	−0.381	0.138
COVID-19 IU	0.445	0.028	15.653	<0.001	0.385	0.505
Pandemic-focused time	0.379	0.137	2.759	0.006	0.089	0.675
Perceived efficacy	−0.231	0.033	−7.089	<0.001	−0.303	−0.160
COVID-19 IU × Perceived efficacy	−0.018	0.008	−2.652	0.008	−0.034	−0.002
**Indirect effect**			*Effect*	*Boot SE*	*Boot LLCI*	*Boot ULCI*
COVID-19 IU → Pandemic-focused time → Negative emotions			0.138	0.006	0.003	0.027
**Conditional direct effect**			*Effect*	*Boot SE*	*LLCI*	*ULCI*
M − 1 SD			0.515	0.042	0.432	0.598
M			0.445	0.028	0.389	0.500
M + 1 SD			0.374	0.035	0.305	0.443

Note. Bootstrap sample size = 5000. LL = low limit, UL = upper limit, CI = confidence interval.

## Data Availability

The data that support the findings of this study are available from the corresponding authors, Q.L. and B.D., upon reasonable request.

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
