# Peer review of "The Impact of Intolerance of Uncertainty on Negative Emotions in COVID-19: Mediation by Pandemic-Focused Time and Moderation by Perceived Efficacy"

_ijerph, 2021, doi:10.3390/ijerph18084189_

Round 1
Reviewer 1 Report
I thank the authors for their very careful and full response to comments of both reviewers. The paper is, as a consequence, a much stronger paper and is a useful contribution to the literature.
Reviewer 2 Report
The authors have responded adequately to the indications
This manuscript is a resubmission of an earlier submission. The following is a list of the peer review reports and author responses from that submission.
Round 1
Reviewer 1 Report
This is a well researched and well referenced paper. My main concern is readability - it is very terse and in places feels repetitive - but the English is fine grammatically with no typos (that this reviewer spotted anyway). Overall the paper is investigating a topic of high interest - and has relevance for the study of anxieties beyond the covid-19 context. The final sentence of the first para in the introduction expresses this very well. The model and methodology are well described.
The first paragraph of the introduction (lines 22-23) requires a specific date for these data to identify the meaning of "now" in line 21 - the pandemic was still in full swing at that stage and subsequent re-emergences and variant strains, plus the emergence of a genuine second wave after this (and the ongoing risk of a third wave), means that by the time readers see this paper the numbers will be much higher.
Would it be possible to append the questionnaire used? This would be useful in assessing the likely biases in the data (noting that the section on limitations is an excellent analysis). For example the collection method ensures that there are implicit biases towards the better educated people who (may) have quite different interactions between anxieties from the less educated, in addition to differences in age/sex distributions. Related to this - was there any indication that the field of expertise among the educated group affected outcomes? A science graduate may have a quite different outcome from a humanities graduate. In particular anybody working in public health or virology would obviously spend much more time looking at information on covid - the real point of interest would be the obsessiveness of the behaviour. Lines 272-275 is an extremely important point, which might reward further investigation as a wider aspect of psychology (especially in conjunction with perceived efficacy)
I do find the work on perceived efficacy both interesting and one of the strengths of this study. Overall a useful paper - my main comment is that the limitations of the study are well recognised and expressed - and this makes the paper useful as a springboard for further research.
Reviewer 2 Report
The topic is very topical and interesting and the number of questionnaires is very high, but reading the document has given me some doubts:
On lines 124-130, the assessment of depression, anxiety and fear is indicated. They are not made through any questionnaire or validated questionnaire items. A single question to determine the intensity of depression that the subject has experienced in the last 10 days seems very scarce.
I therefore recommend not publishing it in the journal, as it does not meet the minimum standards for a journal indexed in such an important quartile in JCR.